# Treatment of Achilles Tendinopathy in Recreational Runners with Peritendinous Hyaluronic Acid Injections: A Viscoelastometric, Functional, and Biochemical Pilot Study

**DOI:** 10.3390/jcm10071397

**Published:** 2021-03-31

**Authors:** Marco Gervasi, Elena Barbieri, Italo Capparucci, Giosuè Annibalini, Davide Sisti, Stefano Amatori, Vittoria Carrabs, Giacomo Valli, Sabrina Donati Zeppa, Marco Bruno Luigi Rocchi, Vilberto Stocchi, Piero Sestili

**Affiliations:** 1Department of Biomolecular Sciences, University Urbino Carlo Bo, via A. Saffi 2, 61029 Urbino, Italy; elena.barbieri@uniurb.it (E.B.); italo.capparucci@libero.it (I.C.); giosue.annibalini@uniurb.it (G.A.); davide.sisti@uniurb.it (D.S.); s.amatori1@campus.uniurb.it (S.A.); vittoria.carrabs@uniurb.it (V.C.); giacomo.valli@uniurb.it (G.V.); sabrina.zeppa@uniurb.it (S.D.Z.); marco.rocchi@uniurb.it (M.B.L.R.); vilberto.stocchi@uniurb.it (V.S.); piero.sestili@uniurb.it (P.S.); 2Interuniversity Institute of Myology (IIM), 06121 Perugia, Italy

**Keywords:** tendinopathy, Achilles tendon, hyaluronic acid, viscoelastic properties, isometric contraction, matrix metalloproteinase 3, interleukin-1beta

## Abstract

Background: Achilles tendinopathy (AT) affects ca. 10 million recreational runners in Europe; the practice of hyaluronic acid (HA) infiltration is being increasingly adopted. The aim of this pilot study was to monitor the effects of a three-local time-spaced injections regimen of HA in the treatment of AT in middle-aged runners combining for the first time viscoelastometric, biochemical, and functional methodologies with routine clinical examinations. Methods: Eight male runners (Age 49.3 ± 3.9), diagnosed for unilateral AT, were given three ultrasound (US) guided peritendinous HA injections at the baseline (T0) and every fifteenth day with a follow-up on the forty-fifth day (T1, T2, and T3). At all-time points patients were assessed for viscoelastic tone and stiffness, maximal voluntary isometric contraction (MVIC), and pain level (Likert scale 0–5). The peritendinous effusions of the injured tendon were collected at T0 and T2 to quantify the volume variations and the IL-1β and MMP-3 levels. Results: At T0 MVIC and pain score were significantly lower and higher, respectively, in injured tendons. The volume, IL-1β and MMP-3 levels decreased in the course of treatment and the clinical endpoints ameliorated over time. Tone, stiffness, and functional performance also varied significantly at T2 and T3, as compared to T0. Conclusions: The sequential peritendinous injections of HA were effective in the amelioration of the clinical symptoms, as well as of the functional and viscoelastic state associated with AT. The determination of the viscoelastometric state may help to precisely evaluate the healing process in AT patients.

## 1. Introduction

Tendinopathies are degenerative musculoskeletal conditions occurring across the age spectrum. It accounts for up to 30% of general practice musculoskeletal consultations, mostly in active and sporting people [1]. Notably, overuse tendon injury is a condition where a tendon has been repeatedly strained until it is unable to withstand further loading, at which point damage occurs, and it is claimed to account for 30–50% of all sports-related injuries [2].

Achilles tendons are the most common site of injuries accounting for 6–17% of all sports-related injuries [2]. Achilles tendinopathies (AT) represent serious injuries for athletes of all levels, often causing (up to 5%) the end of a career in professional sports [3]. In runners, the Achilles tendon musculotendinous unit, consisting of the fusion of the gastrocnemius and soleus tendons, provides the primary propulsive force for locomotion. During running, the Achilles tendon load reaches six to eight times the bodyweight, a load close to the ultimate strength of the tendon [4]. Training errors, including a sudden increase in training volume and/or intensity, changing of terrain or shoe, or an excess in interval training, are the most common causes of Achilles tendon damage of runners [5].

The term “Achilles tendinitis” implies an inflammatory pathologic process within the tendon itself. There are many terms given for the same type of pathologic entity denoting inflammation of the paratenon, such as tenosynovitis, tenovaginitis, peritendinitis, or paratenonitis. Furthermore, various pathologic conditions sometimes coexist (for example, paratenonitis with tendinosis). However, in all cases, the tendinopathy is a failed healing response, with degeneration and haphazard proliferation of tenocytes, disruption of collagen fibers, and subsequent increase in the non-collagenous matrix [6]. In these processes, it has been reported that pro-inflammatory cytokines play a major role [7]. In particular, the interleukin-1 beta (IL-1β) regulates inflammatory mediators and matrix metalloproteinases (MMPs) which degrade the extracellular matrix (ECM) and contribute to the development of tendinopathy and even tendon rupture [8].

The first suspicion of AT is based on history and clinical examination of the patient [9]. Patients commonly experience morning stiffness after a period of inactivity, followed by a gradual onset of pain during activity; in severe cases, pain occurs also at rest [9]. Diagnosis can be integrated by magnetic resonance imaging (MRI) or Ultrasound imaging (USI). Nowadays, it is also possible to monitor a number of mechanical and functional parameters, which may not only integrate clinical assessments, but also add novel and valuable information on the progression of AT and the efficacy of treatments. USI has been used to evaluate the thickness and cross-sectional area (CSA) in musculoskeletal conditions including AT. In particular several studies, [10,11,12,13] demonstrated that the etiology of AT is multifactorial, showing changes in the thickness and CSA of the tendons and of intrinsic and extrinsic foot muscles. These alterations are of paramount importance, as it will be discussed, in the sports medicine management of athletes. To this regard, a novel approach based on handheld myotonometer, a non-invasive digital palpation device, has the capability to precisely determine quantitative differences in viscoelastometric parameters, specifically the transverse stiffness and tone of Achilles tendon [14]. These parameters reflect the tendon structural state, which conceivably varies as a result of AT. Indeed, it has been demonstrated that the symptomatic AT tissue is softer (more compliant) in the painful region [15]; moreover, Finnamore et al. [16] found a lower stiffness in AT as compared with the healthy tendon in recreational runners. These findings suggest that the AT pathological events impact on structural/viscoelastic properties of the tendon. Another functional parameter that can be monitored in AT is the maximum voluntary isometric contraction (MVIC) of the ankle plantar flexion; indeed, people with AT may display reduced maximal plantar flexor torque due to the tenderness and the reduced ability of the tendon to transfer forces to the joint and/or to the onset/presence of pain [17].

The standard pharmacological treatment of AT typically involves systemic nonsteroidal anti-inflammatory drugs (NSAIDs) without or with local corticosteroid injections [18], whose frequent and repeated use may however increase the risk of tendon rupture [19].

Despite very few studies on AT in humans, among the therapeutic options for tendinopathies peritendinous hyaluronic acid (HA) injection is gaining increasing importance as a reliable option for the management of this disease. Under homeostatic conditions, HA is a high molecular mass polymer that is naturally found in most of the tissues, particularly in the ECM of soft connective tissues and synovial fluids of vertebrates. HA regulates important physiological processes related to tissue integrity [20]: it possesses unique viscoelastic properties, is an ideal biological lubricant and also exerts analgesic, anti-inflammatory, and anti-adhesive effects [21]. Recently, clinical trials showed the efficacy and safety of treatment with a cycle of three low molecular weight HA peritendinous injection (one per week, 2 mL, 500–730 kDa) on pain reduction in patients affected by lateral elbow, Achilles, or patellar tendinopathy [22,23]. Another recent study, although on a different human tendinopathy setting (long head of biceps tendinopathy), also showed that high molecular weight (3000 kDa) HA treatments decrease the inflammatory marker levels in the peritendinous effusion and ameliorate tendinopathy-associated symptoms [24].

That being the case, we hypothesize that the sequential tendon infiltration of HA can reduce the inflammation and improve the clinical and functional parameters. Hence, this pilot study aimed to better evaluate the efficacy of treatment with a cycle of three peritendinous injections of 2–1000 kDa HA (one each 15-day, 2 mL, RegenFlex T&M, Regenyal Laboratories SRL, Italy) on clinical, viscoelastometric, functional, and biochemical determinations, in middle-aged recreational runners affected of unilateral AT.

## 2. Materials and Methods

We performed a pilot study carried out in a real-life clinical setting on recreational runners diagnosed with unilateral AT and to whom the specialist physician prescribed three local injections of HA every fifteen days with a follow-up visit to the forty-fifth day. After approval from the institutional ethical committee on 26 November 2018, the study was carried out according to the Helsinki Declaration for research with human volunteers. All patients were enrolled from 10 May 2019 to 30 June 2019 and signed an informed consent form to participate.

### 2.1. Participants

Eight male patients were enrolled (age 49.3 ± 3.9; weight 81.1 ± 15.0 kg; height 173.3 ± 10.3 cm; BMI 27.1 ± 5.0 kg/m^2^). The inclusion criteria were an experience of at least 4 years in recreational running; pain with tendinopathic features and peritendinous effusion on US imaging before HA treatments.

The exclusion criteria were suspected tendon rupture or insertional tendinopathy, general, severe inflammatory-based illnesses (diabetes mellitus, rheumatoid arthritis, peripheral neuropathy), known sensitivity to HA, smokers, and patients with BMI ≥ 35. Subjects taking supplements (i.e., chondroitin sulphate or methylsulfonylmethane) or medications, including steroidal and nonsteroidal anti-inflammatory drugs per os or per infiltration in the target tendon within the last 3 months were also excluded.

Patients were allowed to walk immediately after the infiltration, but were advised to refrain from running and any other type of moderate/vigorous activity for three to four days after the first injection and at least within forty-eight hours following the second and the third injections. Moreover, any kind of physiotherapy or rehabilitation exercises were not permitted as well as the use of any type of orthosis for the entire study duration.

### 2.2. Experimental Design

Viscoelatometric and functional parameters were chosen to obtain qualitative and quantitative information on the progression of AT and the efficacy of treatments. All patients were given three (US) guided peritendinous injections (every fifteenth day) of HA (molecular weight, MW): a blend of 2 to 1000 KDa, 2 mL (RegenFlex T&M, Regenyal Laboratories SRL, Italy) according to the methodology described by Frizziero et al. [23]. US evaluations were performed by an experienced ecographist in the clinic center and no anesthetic or rescue drug was used after injections. The patients underwent the examinations lying prone with the foot hanging freely over the edge of the examination table to inspect the Achilles tendon. Tendinopathy with peritendinous effusion was confirmed using US imaging and a sonographic transducer. A 1.5-inch needle was visualized in the long axis of the transducer and, with US control, was advanced. Once the needle tip was seen within the peritendinous effusion, the effusion was aspirated and collected for further analysis as described in Wu et al. [24], Chiodo et al. [25], and Peters et al. [26]. HA was injected into the peritendinous area using real-time US monitoring (T0). Thirty days after the first HA injection (T2), the AT peritendinous effusion was confirmed, aspirated, and collected again for further analysis. A clinical evaluation of tendinopathy based on redness, warmth, swelling, tenderness, and crepitus during movement, peritendinous effusion was performed at any clinical visit and adverse events were assessed for safety. No adverse effect was observed. The contralateral non-painful tendon was also examined as an intra-patient comparison. Before each HA injection at the specific time points (T0, T1, T2, and T3), MVIC parameters and the level of the patient’s pain were respectively assessed (Figure 1).

### 2.3. Clinical and Functional Assessments

On the first day of visit (T0), weight and height were measured, and the body mass index was calculated; the clinical examination and magnetic resonance imaging (MRI) were performed to confirm the diagnosis or to exclude severe tendon lesions. The criteria for a diagnosis of tendinopathy were primarily based on patient history and physical examination based on Royal London Hospital Test [9]: in this test, the medical staff elicits local tenderness by palpating the tendon with the ankle in neutral position or slightly plantar flexed. The tenderness significantly decreased or became totally painless when the ankle was dorsiflexed. With the ankle in maximum dorsiflexion and in maximum plantar flexion, the portion of the tendon originally found to be tender was palpated again. Results were classified as presence or absence of tenderness on dorsiflexion. In asymptomatic tendons, the test was performed selecting an area in the tendon 3 cm proximal to its calcaneal insertion when the ankle was held in neutral as described by Maffulli et al. [9].

In order to determine the viscoelastic state of the tendon, a handheld myotonometer (MyotonPro; Myoton Ltd., Tallinn, Estonia) capable of measuring the quantitative mechanical parameters of the Achilles tendon in vivo was used. This device has good to excellent test-retest reliability and has been established in previous studies in muscles and tendons [21,22]. The assessments were carried out by an expert in the use of myotonometer technology. Participants were asked to lay prone on a couch; both injured and healthy tendon of each participant were then assayed positioning the probe of the device (3 mm in diameter) along the tendon, at 8 cm from the plantar aspect of the heel up to the proximal component of the musculotendinous junction [21]. Tendons’ tone and stiffness were measured at the relaxed state. The probe of the device, preloaded at 0.18 N, applied brief (15 ms) low force (0.4 N) mechanical impulses, inducing damped natural oscillations of the underlying tendon. These oscillations were recorded by an accelerometer connected to a friction measurement mechanism in the device. The device then simultaneously calculated the resting tone [frequency of oscillation (Hz)] and stiffness (N/m) parameters. A set of 10 mechanical impulses at one-second intervals was carried out and the mean of each set of 10 was used for analysis. To determine the MVIC participants performed three repetitions of a MVIC of a plantar flexion according to the similar procedure applied by Kelly et al. [23]. Subjects were asked to lay down over a couch in the prone position. Upper and lower back, as well as the popliteal fossa were anchored to the couch with some straps. Another band was wrapped at the metatarsal level, connected to a ring and a carabiner; the band was covered and tape-fixed on the foot to avoid accidental movements and the ankle was maintained at 0 degrees neutral (see Figure 2).

Then, a steel cable was connected to the carabiner and anchored to a fixed platform on the floor to the opposite side of the couch where the patient’s head rested. A load cell connected between the steel cable and the platform measured the force produced by the subject during the isometric gastrocnemius contraction. In order to obtain an isometric contraction, before each trial, the cable was pre-loaded with 1 Kg tension. Each subject performed three familiarization trials and three maximal trials for each foot, with a 3-min recovery in between. At the end of each trial (both familiarization and maximal), pain intensity scale was shown to the subjects and measured with a 5-point Likert scale (0 = no pain, 1 = low pain intensity, 2 = medium pain intensity, 3 = high pain intensity, and 4 = severe pain intensity) [24]. Both for the MVIC and the pain scale, the highest scores between the three maximal trials were considered for the analysis.

### 2.4. Biochemical Assessment

To support the clinical and functional evaluation of the HA treatment, we also analyzed specific pro-inflammatory mediators in the peritendinous effusion aspirated after HA treatment. Quantification of IL-1β and MMP-3 levels was performed using Enzyme-linked Immunosorbent assay (ELISA) test. Human IL-1 beta/IL-1F2 (HSLB00D) and Human Total MMP-3 (DMP300) Quantikine High Sensitivity ELISA Kits were used (R&D Systems, Milan, Italy). The results were detected at a wavelength of 450 nm using a spectrophotometer reader.

### 2.5. Statistical Analyses

Descriptive statistics of variables considered were performed reporting means and standard deviations at different time measurements. In order to verify changes during time in treated and untreated arms, two-way mixed designs (MANOVA for repeated measures) were performed. Time was within-subjects 4 levels factor (T0, T1, T2, T3). Unilateral tendon (treated-injured vs. untreated-healthy condition) was 2 levels between group’s factors; age and BMI were covariates; pain level, MVIC, frequency, and stiffness were dependent variables. Contrasts are used to test for differences among the levels of a between-subjects factor; difference contrast compares the mean of each level (T1, T2, T3) to the previous level. IL-1β and MMP3 levels at T0 and T2 were compared using MANOVA for repeated measures. Overall and partial Eta squared was used as effect size estimation. When epsilon was >0.75, the Huynh–Feldt correction was applied and when epsilon was <0.75, the Greenhouse-Geisser correction was applied. All elaborations were conducted with alpha = 0.05. Elaborations and graphics were obtained using Excel 365 (Microsoft) and SPSS version 20.0 (SPSS Inc., Chicago, IL, USA).

## 3. Results

Clinical exams confirmed the diagnosis of AT, while MRI excluded severe lesions or tendon ruptures. The treatment proved to be safe and well tolerated, and no adverse effect was observed. Repeated measures MANOVA test was conducted to test injection effects on pain, MVIC, and viscoelastometric parameters. Mauchly’s Sphericity test was not significant for pain, frequency, and stiffness, while MVIC was significant (*p* = 0.006); for the latter variables, Huynh-Feldt or Greenhouse-Geisser correction was applied. The results showed that during overall time (T0–T3), pain, MVIC, frequency, and stiffness were significantly different between injured and contralateral tendon (*p* = 0.028; *p* = 0.023; *p* = 0.047, respectively); this is mainly due to the difference in T0; in later times the differences become very small. The results showed that during time, pain, MVIC, frequency, and stiffness were significantly different between healthy and injured tendon (*p* = 0.028; *p* = 0.023; *p* = 0.047, respectively). Considering the age and BMI as covariates, only frequency was related to these parameters (*p* = 0.002 and *p* = 0.001, respectively), while pain, MVIC, and stiffness were independent of age (see Table 1).

Contrast analysis of differences highlighted when injured tendons got better until a non-significant difference in respect to the contralateral one was reached. The pain was higher in the injured tendon at T0; ΔT1-T0 contrast was significant (*p* = 0.042; η^2^ = 0.39, see Figure 3a); T1-T0 decrease of pain in injured tendon was high and reached the average value of normal tendon; consequently, ΔT2-T1 and ΔT3-T2 were not significant (*p* > 0.05; η^2^ = 0.13). Frequency showed a similar pattern: ΔT1-T0 showed a rise of frequency in injured tendons (*p* = 0.048; η^2^ = 0.40), while there were not significant variations in the following time points (see Figure 3c). A post-hoc analysis of power was performed for stiffness and tone variables; a difference between two dependent means (matched pairs, T0-T1 interval) was considered. Stiffness variable, in T0, was 872 ± 29.1; in T1 it was 935 ± 35.6 for an effect size (Cohen d) = 1.94. Considering alpha = 0.05 and a two tailed *t* test for paired data, with eight subjects, we reached a power (1-beta) = 0.99. Tone variable, in T0, was 32.2 ± 0.77; in T1 it was 33.3 ± 0.75 for an effect size (Cohen d) = 1.05. Considering alpha = 0.05 and a two tailed *t* test for paired data, with eight subjects, we reached a power (1-beta) = 0.78.

Similarly, MVIC (*p* = 0.004; η^2^ = 0.67) and stiffness (*p* = 0.035; η^2^ = 0.44) showed similar patterns (see Figure 3b,d). In brief, the first injection seems to account for the majority of the observed beneficial effects and the subsequent injections to maintain the improvements achieved. The volume of peritendinous effusion significantly dropped by about 57% at T2 (from 420.00 ± 40 uL to 238.75 ± 25 uL; *p* = 0.018; η^2^ = 0.63) and was associated with a reduction of IL1-β and MMP-3 levels (*p* = 0.027; η^2^ = 0.58) (Figure 4a,b).

## 4. Discussion

This pilot study describes for the first time the effect efficacy of a cycle of three HA peritendinous injections over a forty-five-day period on clinical, viscoelastometric, functional, and biochemical parameters, in middle-aged recreational runners affected by unilateral AT.

Our data indicate that this treatment promoted a significant and rapid amelioration of the AT-associated alterations of the above parameters in all the patients. Relief from clinical symptoms such as tenderness on palpation and pain was very rapid; accordingly, peritendinous effusion volume reduction was observed after the treatment; the biochemical markers of inflammation significantly ameliorated at the selected checkpoints.

In addition, this pilot study is the first to adopt an experimental design combining the above determinations with tendon’s viscoelastic—namely, tone and stiffness—and functional assessments over the course of the HA infiltrative treatment. In this regard, tone and stiffness are two parameters expressing the viscoelastic state of living tissues (muscles and tendons), which have been shown to reflect their biomechanical/functional status and integrity [14,27]. The tone and stiffness values augment with the increase of the contraction levels, exhaustive activity, and ageing [28,29]. Recently Morgan et al. [30] and Finnamore et al. [16] reported that AT induces appreciable alterations in Achilles tendon viscoelasticity suggesting a correlation between tendon integrity and its viscoelastic properties.

Here we report that all the considered parameters significantly varied over the treatment period in injured AT, reaching values similar or even superimposable to those of the contralateral healthy one. AT also resulted in functional impairment, as indicated by the decrease of MVIC in the homolateral limb. Interestingly, fifteen days after the first injection, all these parameters changed. It is worth considering that this implies that the functional and viscoelastometric asymmetries between the two limbs characterizing the pre-treatment stage tended to decrease and then disappear. In particular, the MVIC and the tone assessed in the AT tendon increased until they reached values similar and not statistically different from the healthy one. As to the stiffness values, fifteen days after the first injection, they equilibrated as well; however, in this case, it is worth noting that this achievement depended not only on the increase in AT tendon stiffness, but also on the concomitant slight reduction occurring in the healthy tendon. Concerning this, a likely hypothesis to explain this result may be that, in the absence of treatments, the healthy tendon stiffness increases in response to the altered/excessive distribution of loads obligatorily associated with the AT which impacts on the contralateral limb biomechanical efficiency. The equilibrium in tone and stiffness between the healthy and inflamed tendon did not vary following the two injections at T2 and T3, while the MVIC increased constantly and symmetrically on both sides. In parallel with the balancing of the viscoelastic and functional parameters, the perceived pain drastically dropped after the first injection to the same values of the healthy tendon and did not vary up to the last clinical assessment.

Hence, these parameters may help, especially in runners, to accurately identify when the two limbs equilibrate each other and the athlete can return to normal activity, limiting the risk of underestimating the residual injury and of AT recurrence; however, further research is needed to generate basic data for specific population groups that monitor these variables over time.

In line with the clinical and functional amelioration due to the HA treatment, we also observed a significant reduction of specific pro-inflammatory mediators in the peritendinous effusion aspirated after HA treatment. Indeed, IL-1β, which is recognized as the initiator of tendinopathy since it induces inflammation, apoptosis, and ECM degradation by activating MMPs, drastically dropped out after HA treatment. In line with the fall of IL-1β and according to Del Buono et al. [31], we also found a significant reduction of MMP-3 levels, the endopeptidase that degrades the ECM and mediates the development of tendinopathy. Our data are in agreement with the finding of Wu et al. [24], who showed that high-molecular-weight HA attenuated tendinopathy by down regulating MMP-1 and -3 expression via CD44. Indeed, HA, beyond its physical and lubricating properties, is actively involved in regulating inflammatory responses mediated through the interactions with different substrates and receptors, among which CD44 represents a primary target [32] and has HA also effectively mitigated chondropathy and tendinopathy in clinical practice [33].

The HA utilized in this pilot study is a blend of different MW linear-HA similar in composition to that exhibiting a strong and prompt anti-inflammatory activity in a previous study on osteoarthrosis (OA) from a previous study of our group [33]. Although completely different from each other, a common trait linking the two pathologies is the need to reduce the inflammatory process and its consequences, an effect that both the blends of HA proved to afford. Indeed, in strict analogy with previous data on OA, here, we found that the AT-inflammatory hallmarks IL-1β and MMP-3 were rapidly and strongly reduced immediately after the first injection. Differently from OA, where very high MW HA is required—and usually included in injectable preparations—to afford viscosupplementation within the joint [33], in AT is unneeded. Rather, it is important that HA has some chance to diffuse around and through the paratenon sheath to reach the surrounding inflammation sites, a feature which is inversely related to MW; secondarily, some positive mechanical effect may derive from the lubrication of injured tendon, a property which is retained by the heavier HA fraction around 1.000 KDa.

Taken together these results confirm that the treatment with peritendinous HA injections allows a reduction of the main symptoms characterizing AT and allows the patient to resume the main basic functions. These data are in agreement with the recent clinical evidence [22,23,24]. Here, our pilot study shows that AT is accompanied by alterations of stiffness and tone of the tendon body, supporting the concept that tissue mechanical properties are a marker of disease. Since stiffness and tone can be precisely and reliably determined, their measurement provides the clinician with a simple, rapid, and non-invasive method to objectively quantitate the extent of tendon recovery.

## 5. Strengths and Weaknesses of the Study

The strength of the present study, as compared to previous ones on the same or similar topics [16,17,24] is that it is the first one adopting a multi-methodological approach in a longitudinal setting. Indeed, this approach—that could be extended to larger clinical studies—allows to gather an integrated, complex data set accurately and objectively reflecting the quali-quantitative state of the AT changes induced by HA treatment. Furthermore, we highlight the capacity of viscoelastometry to easily, rapidly, and non-invasively assess the state of AT with the additional advantage of providing quantitative parameters.

Conversely, the main limitations were the small number of the enrolled sample and the lack of a control group of AT runners treated according to the standard of care. Although this limitation is not uncommon in similar studies focusing on “real life” settings, it is of worth that the aim of this pilot study was to set the basis for future, larger clinical studies, rather than comparing the efficacy of HA with the standard of care. Moreover, another limitation is that the patients enrolled in this study—the recreational runners—while likely representative of sport practitioners and athletes suffering of AT, might not necessarily reflect the situation of other AT population subgroups, i.e., sedentary and/or elderly and/or overweight people. Again, future clinical studies should include these population subgroups as well as a standard of care group.

## 6. Conclusions

In conclusion, our results point out the therapeutic potential of low to medium MW HA in treating one of the most diffused tendon pathologies. In particular, we noticed a progressive improvement of all the tested parameters over the treatment with HA, leading to a significant reduction of functional and mechanical asymmetries between AT and Healthy limbs. Furthermore, this study proposes a new multi-methodological and integrated approach which might pave the way to larger clinical studies focusing on the pharmacological treatment of AT, a very common and subtle condition.

## Figures and Tables

**Figure 1 jcm-10-01397-f001:**
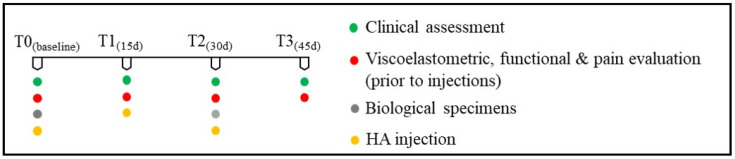
Experimental design.

**Figure 2 jcm-10-01397-f002:**
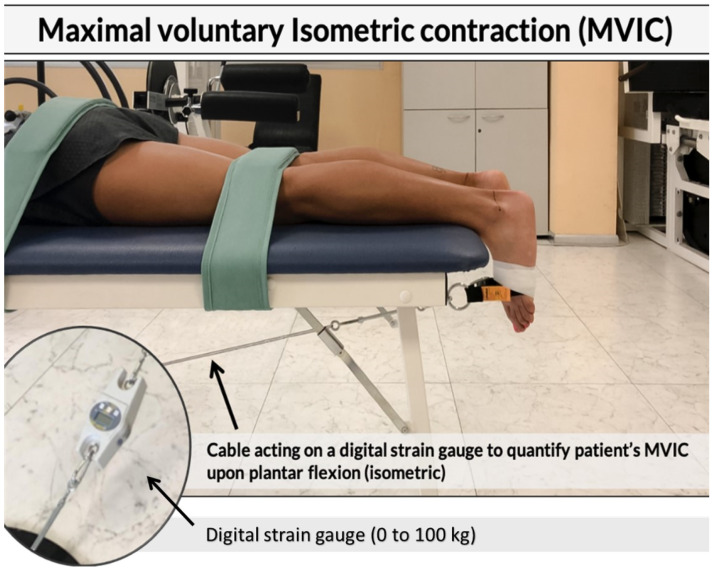
Subject positioning for isometric gastrocnemius contraction in prone. The participant was prone, shoes off with feet hanging unsupported off edge of the couch, and knees resting in 0 degrees extension. The trunk and lower extremities were anchored to the table by a strap just above the popliteal crease and across the pelvis at the level of the greater trochanters. The ankle position was maintained at 0 degrees neutral.

**Figure 3 jcm-10-01397-f003:**
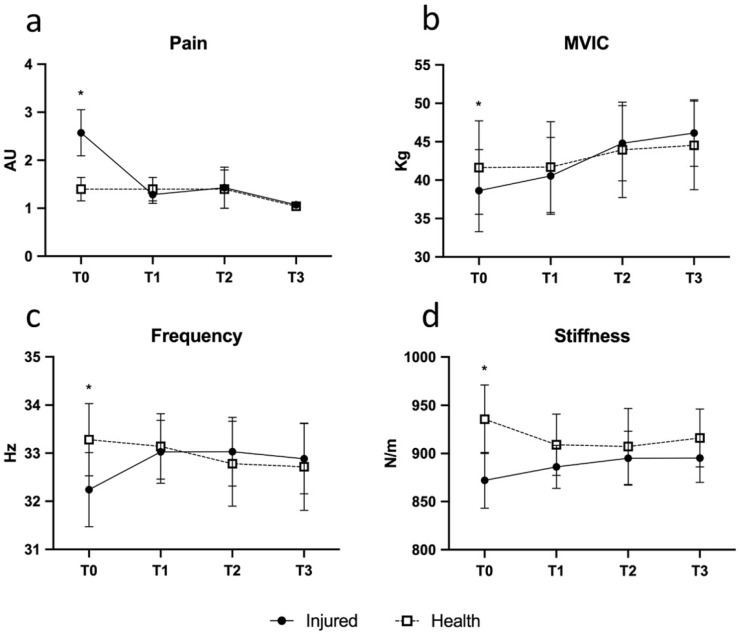
Time course variations of pain assessed with a 5-point Likert scale (**a**) (0 = no pain, 1 = low pain intensity, 2 = medium pain intensity, 3 = high pain intensity, and 4 = severe pain intensity), MVIC (**b**), frequency (**c**), and stiffness (**d**) in the Achilles tendinopathy (AT) (black dots) and contralateral healthy (white dots) tendon. Bars represent standard errors. * *p* < 0.05, intergroup comparison.

**Figure 4 jcm-10-01397-f004:**
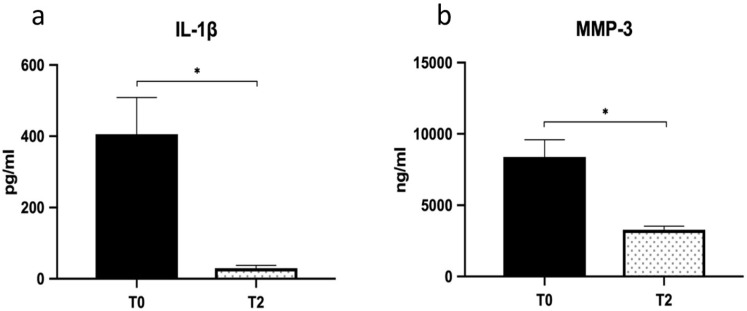
Peritendinous effusion levels of inflammation markers. IL-1β (**a**) and MMP-3 (**b**) levels were determined in peritendinous effusion at baseline (T0) and at one month (T2). Values are means ± SE. *, significantly different as compared to baseline (*p* < 0.05).

**Table 1 jcm-10-01397-t001:** Functional and viscoelastometric measurements.

	T0		T1		T2		T3	
Variables	Injured	Healthy	Injured	Healthy	Injured	Healthy	Injured	Healthy
Pain (AU)	2.57 (0.48)	1.40 (0.24)	1.29 (0.18)	1.40 (0.25)	1.43 (0.43)	1.40 (0.40)	1.07 (0.07)	1.04 (0.04)
MVIC (kg)	38.6 (5.35)	41.6 (6.08)	40.5 (5.01)	41.7 (5.91)	44.8 (4.90)	43.9 (6.22)	46.1 (4.34)	44.5 (5.78)
Tone (Hz)	32.2 (0.77)	33.3 (0.75)	33.0 (0.65)	33.1 (0.68)	33.0 (0.71)	32.8 (0.89)	32.9 (0.73)	32.7 (0.91)
Stiffness (N/m)	872 (29.1)	935 (35.6)	886 (22.4)	909 (31.9)	895 (28.1)	907 (39.6)	895 (25.3)	916 (30.1)

Mean (standard deviation) of pain, maximal voluntary isometric contraction (MVIC), frequency, and stiffness measured on injure and healthy tendons at different times: T0 (baseline), T1 (15d), T2 (30d), and T3 (45d).

## Data Availability

The data presented in this study are available on request from the corresponding author. The data are not publicly available for ethical and privacy reasons.

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
