# Peer review of "Treatment of Achilles Tendinopathy in Recreational Runners with Peritendinous Hyaluronic Acid Injections: A Viscoelastometric, Functional, and Biochemical Pilot Study"

_jcm, 2021, doi:10.3390/jcm10071397_

Round 1

Reviewer 1 Report

Thank you for opportunity for reviewing this interesting paper. The research adhere to reporting observational guidelines. After carefully reading this manuscript, I must say that, from my point of view, the authors have done research on an important topic Peritendinous hyaluronic acid injections on Achilles tendinopathy in recreational runners: a viscoelastometric, functional and biochemical pilot study. This could be interesting clinicians, universities, private research organizations, and independent scientists, that frequently work in this area.  It could give them a wider concept about and helps advance recognition of the input of different health professionals into the management of this condition, and helps inform the need for further multi-professional work in this area. This is an interesting aim with the Peritendinous hyaluronic acid injections on Achilles tendinopathy in recreational runners: a viscoelastometric, functional and biochemical pilot study. I have considered the quality of the manuscript redaction and presentation, the quality of the research methodology, the novelty and importance of the observations, and the appropriateness for the Journal's readers (according with the Journal's name) and I think that this manuscript joins adequate conditions to be accepted for publication Journal of Clinical Medicine.
I have no real problems with the text of this paper, only some suggestions that are mentioned below. It appears as if the authors have done the study well and have answered an interesting clinical question with their work.
Also, there are a major concerns with the manuscript that require attention prior to publication. These will be discussed below relative to the sections of the manuscript.

TITLE
The title should be amended slightly to ensure that the reader understands immediately to type of research realized for analize  expect problems in Achilles tendinopathy.
KEYWORDS:
Please use recognised MeSH terms as this will assist others when they are searching for information on your research topic. The following website will provide these (simply start typing in a keyword and see if it exists or find an alternative if it does not): https://www.ncbi.nlm.nih.gov/mesh
INTRODUCTION
I suggest that background should be improved, with more details about  the importance Achilles tendinopathy. It is indeed important paper but it lacks several critical references, in which it was presented related with this condition, and it should be emphasized in the INTRODUCTION or Discussion of the authors' paper. More info info in

Ultrasonography Features of the Plantar Fascia Complex in Patients
with Chronic Non-Insertional Achilles Tendinopathy: A Case-Control Study
https://pubmed.ncbi.nlm.nih.gov/31052554/

Intrinsic foot muscles morphological modifications in patients with
Achilles tendinopathy: A novel case-control research study
https://pubmed.ncbi.nlm.nih.gov/31593918/

Ultrasound evaluation of extrinsic foot muscles in patients with
chronic non-insertional Achilles tendinopathy: A case-control study
https://pubmed.ncbi.nlm.nih.gov/30844628/

Also, please include hypothesis in this section

METHODS
This section is poor, needs to present a better rationale for the study and the methodology employed. Also, neither appear information related with inclusion and exclusion criteria, dates, protocol.
Likewise more detail about information calculate sample size and data should be provided. Also, please need include the data and record code and all information related with the ethics committee and explain aspects ethics and legal requirement about this research.
RESULTS The results is clear and concise with appropriate statistical analysis been performed appropriately and rigorously.
DISCUSSION Include this section the principal strengths and weaknesses in relation to other studies, discussing important differences in results; the meaning of the study: possible explanations and implications and unanswered questions and future research
CONCLUSION: Summarize the conclusions in order to reflect only the study findings. 
TABLES AND FIGURES:  Correct.

Author Response

We carefully took into consideration all reviewers’ comments and we hope that this revised version of the manuscript could be considered suitable for JCM.

All Authors have approved the content and authorship of the revised manuscript.

Reviewer 1)

Comments and Suggestions for Authors

Thank you for opportunity for reviewing this interesting paper. The research adhere to reporting observational guidelines. After carefully reading this manuscript, I must say that, from my point of view, the authors have done research on an important topic Peritendinous hyaluronic acid injections on Achilles tendinopathy in recreational runners: a viscoelastometric, functional and biochemical pilot study. This could be interesting clinicians, universities, private research organizations, and independent scientists, that frequently work in this area.  It could give them a wider concept about and helps advance recognition of the input of different health professionals into the management of this condition, and helps inform the need for further multi-professional work in this area. This is an interesting aim with the Peritendinous hyaluronic acid injections on Achilles tendinopathy in recreational runners: a viscoelastometric, functional and biochemical pilot study. I have considered the quality of the manuscript redaction and presentation, the quality of the research methodology, the novelty and importance of the observations, and the appropriateness for the Journal's readers (according with the Journal's name) and I think that this manuscript joins adequate conditions to be accepted for publication Journal of Clinical Medicine.
I have no real problems with the text of this paper, only some suggestions that are mentioned below. It appears as if the authors have done the study well and have answered an interesting clinical question with their work.
Also, there are a major concerns with the manuscript that require attention prior to publication. These will be discussed below relative to the sections of the manuscript.

TITLE
The title should be amended slightly to ensure that the reader understands immediately to type of research realized for analize expect problems in Achilles tendinopathy. 

  1. The title was modified as following: 

Treatment of Achilles tendinopathy in recreational runners with peritendinous hyaluronic acid injections: a viscoelastometric, functional and biochemical pilot study”

KEYWORDS:
Please use recognised MeSH terms as this will assist others when they are searching for information on your research topic. The following website will provide these (simply start typing in a keyword and see if it exists or find an alternative if it does not): https://www.ncbi.nlm.nih.gov/mesh

  1. The keywords were substituted with the following recognized MeSH terms: 

Tendinopathy; Achilles Tendon; Hyaluronic Acid; viscoelastic properties; Isometric contraction; Matrix Metalloproteinase 3; Interleukin-1beta

INTRODUCTION
I suggest that background should be improved, with more details about  the importance Achilles tendinopathy. It is indeed important paper but it lacks several critical references, in which it was presented related with this condition, and it should be emphasized in the INTRODUCTION (MARCO) or Discussion (PIERO) of the authors' paper. More info info in

Ultrasonography Features of the Plantar Fascia Complex in Patients
with Chronic Non-Insertional Achilles Tendinopathy: A Case-Control Study
https://pubmed.ncbi.nlm.nih.gov/31052554/

Intrinsic foot muscles morphological modifications in patients with
Achilles tendinopathy: A novel case-control research study
https://pubmed.ncbi.nlm.nih.gov/31593918/

Ultrasound evaluation of extrinsic foot muscles in patients with
chronic non-insertional Achilles tendinopathy: A case-control study
https://pubmed.ncbi.nlm.nih.gov/30844628/

  1. We agree with the reviewer in improving the background and adding the suggested references.

Line 66-70: USI has been used to evaluate the thickness and cross-sectional area (CSA) in musculoskeletal conditions including AT. In particular several studies (Cook & Purdam, 2009; Morales et al 2019; Morales et al 2019; Morales et al 2019) demonstrated that the etiology of AT is multifactorial, showing changes in the thickness and CSA of the tendons and of intrinsic and extrinsic foot muscles.

Line 84-86: The standard pharmacological treatment of AT typically involves systemic NSAIDs without or with local corticosteroid injections (Hong-You Li 2016), whose frequent and repeated use may however increase the risk of tendon rupture (Wilson, A. 2018).  

Also, please include hypothesis in this section

We agreed with the reviewer and we added the hypothesis.

Line 101,102: That being the case, we hypothesize that the sequential tendon infiltration of HA can  reduce the inflammation and improve the clinical and functional parameters.

METHODS
This section is poor, needs to present a better rationale for the study and the methodology employed. Also, neither appear information related with inclusion and exclusion criteria, dates, protocol. 

  1. We agree and we better introduce the methodologies employed as it follows: 

2.2. Experimental Design

Line 132,133: Viscoelatometric and functional parameters have been chosen to obtain qualitative and quantitative information on the progression of AT and the efficacy of treatments. 

2.4. Biochemical Assessment 

Line 210-212: In order to support the clinical and functional evaluation of the HA treatment, we also analyzed specific pro-inflammatory mediators in the peritendinous effusion aspirated after HA treatment.

Likewise more detail about information calculate sample size and data should be provided. 

  1. We agree with the reviewer and added a post-hoc power analyses in the results section:

Line 257-263: A post-hoc analysis of power has been performed, for stiffness and Tone variables; a difference between two dependent means (matched pairs, T0-T1 interval) has been considered. Stiffness variable, in T0, was 872±29.1; in T1 was 935±35.6, for an effect size (Cohen d) =1.94. Considering alpha=0.05 and a two tailed t test for paired data, with 8 subjects, we reached a power (1-beta) =0.99. Tone variable, in T0, was 32.2±0.77; in T1 was 33.3±0.75, for an effect size (Cohen d) =1.05. Considering alpha=0.05 and a two tailed t test for paired data, with 8 subjects, we reached a power (1-beta) =0.78.

Also, please need include the data and record code and all information related with the ethics committee and explain aspects ethics and legal requirement about this research. 

  1. These information has been provided in the section ‘Institutional Review Board Statement’. The responsible ethics committee of the ‘San Benedetto Casa di Cura’ Clinic, that evaluated our proposal, does not use an identification code for the approval, but only the date of approval (26/11/2018).

RESULTS The results is clear and concise with appropriate statistical analysis been performed appropriately and rigorously.
DISCUSSION Include this section the principal strengths and weaknesses in relation to other studies, discussing important differences in results; the meaning of the study: possible explanations and implications and unanswered questions and future research 

  1. We agree and added a specific section as it follows:

Line 360-378:

  1. Strengths and weaknesses of the study

The strength of the present study, as compared to previous ones on the same or similar topics (Finnamore, E. et al et al. 2019; McAuliffe, S. et al. 2019; Wu, P.T. et al. 2017) is that it is the first one adopting a multi-methodological approach in a longitudinal setting.  Indeed, this approach - that could be extended to larger clinical studies - allows to gather an integrated, complex data set accurately and objectively reflecting the quali-quantitative state of the AT changes induced by HA treatment.  Also, we highlight the capacity of viscoelastometry to easily, rapidly and non-invasively assess the state of AT with the additional advantage of providing quantitative parameters. Conversely, the main limitations were the small number of the enrolled sample and the lack of a control group of AT runners treated according to the standard of care. Although this limitation is not uncommon in similar studies focusing on “real life” settings, it is of worth that the aim of this pilot study was to set the basis for future, larger clinical studies, rather than comparing the efficacy of HA with the standard of care. Also, another limitation is that the patients enrolled in this study - the recreational runners - while likely representative of sport practitioners and athletes suffering of AT, might not necessarily reflect the situation of other AT population subgroups, i.e. sedentary and/or elderly and/or overweight people.  Again, future clinical studies should include these population subgroups as well as a standard of care group.

CONCLUSION: Summarize the conclusions in order to reflect only the study findings. 

  1. We agree, and we rewrote the conclusions

Line 380-386: In conclusion, our results point out the therapeutic potential of low to medium MW HA in treating one of the most diffused tendon pathologies. In particular we noticed a progressive improvement of all the tested parameters over the treatment with HA, leading to a significant reduction of functional and mechanical asymmetries between AT and Healthy limbs. Also, this study proposes a new multi methodological and integrated approach which might pave the way to larger clinical studies focusing on the pharmacological treatment of AT, a very common and subtle condition.

TABLES AND FIGURES:  Correct.

Reviewer 2 Report

Dear Authors, in my opinion your study is well conducted. Methods and discussion sections are well documented and conclusion are congruent.

1. The main limitation is represented by the small sample size that could impair the statistical analysis and consequently the results.

2. It is not reported if any possible adverse/side effect was collected (i.e. pain/swelling at injection site or others); do you collect adverse effect? in my opinion even no adverse effect was evidenced, you should clearly write on your paper.

3. Functional scale as VISA-A should be considered in your methods.

Author Response

We carefully took into consideration all reviewers’ comments and we hope that this revised version of the manuscript could be considered suitable for JCM.

All Authors have approved the content and authorship of the revised manuscript.

Reviewer 2)

Comments and Suggestions for Authors

Dear Authors, in my opinion your study is well conducted. Methods and discussion sections are well documented and conclusion are congruent.

The main limitation is represented by the small sample size that could impair the statistical analysis and consequently the results.

  1. We agree with the Reviewer and added a specific section including strengths and weaknesses of the study.

Line 360-378:

  1. Strengths and weaknesses of the study

The strength of the present study, as compared to previous ones on the same or similar topics (Finnamore, E. et al et al. 2019; McAuliffe, S. et al. 2019; Wu, P.T. et al. 2017) is that it is the first one adopting a multi-methodological approach in a longitudinal setting. Indeed this approach - that could be extended to larger clinical studies - allows to gather an integrated, complex data set accurately and objectively reflecting the quali-quantitative state of the AT changes induced by HA treatment.  Also, we highlight the capacity of viscoelastometry to easily, rapidly and non-invasively assess the state of AT with the additional advantage of providing quantitative parameters. Conversely, the main limitations were the small number of the enrolled sample and the lack of a control group of AT runners treated according to the standard of care. Although this limitation is not uncommon in similar studies focusing on “real life” settings, it is of worth that the aim of this pilot study was to set the basis for future, larger clinical studies, rather than comparing the efficacy of HA with the standard of care. Also, another limitation is that the patients enrolled in this study - the recreational runners - while likely representative of sport practitioners and athletes suffering of AT, might not necessarily reflect the situation of other AT population subgroups, i.e. sedentary and/or elderly and/or overweight people.  Again, future clinical studies should include these population subgroups as well as a standard of care group.

It is not reported if any possible adverse/side effect was collected (i.e. pain/swelling at injection site or others); do you collect adverse effect? in my opinion even no adverse effect was evidenced, you should clearly write on your paper.

  1. As suggested, we reported the evaluation of possible adverse/side effect in the Experimental design as it follows:

Line 148-151: ‘A clinical evaluation of tendinopathy based on redness, warmth, swelling, tenderness, crepitus during movement and peritendinous effusion was performed at any clinical visit and adverse events were assessed for safety’. 

We also reported the results obtained in the specific Results session.

Line 334,335: ‘The treatment proved to be safe and well tolerated, no adverse effect was observed.’

Functional scale as VISA-A should be considered in your methods. 

  1. Good point. We will adopt the VISA-A questionnaire for the ongoing study.  However, for the present one, the clinician adopted the Royal London Hospital Test.

Round 2

Reviewer 1 Report

I do not have any further comments and/or suggestions to authors. Thank you very much for the effort and implementation of suggestions to the manuscript.